# Spectral Analysis and Mutual Information Estimation of Left and Right Intracardiac Electrograms during Ventricular Fibrillation

**DOI:** 10.3390/s20154162

**Published:** 2020-07-27

**Authors:** Milton Fabricio Pérez-Gutiérrez, Juan José Sánchez-Muñoz, Mayra Erazo-Rodas, Alicia Guerrero-Curieses, Estrella Everss, Aurelio Quesada-Dorador, Ricardo Ruiz-Granell, Alicia Ibáñez-Criado, Alex Bellver-Navarro, José Luis Rojo-Álvarez, Arcadi García-Alberola

**Affiliations:** 1Departamento de Eléctrica y Electrónica, Universidad de las Fuerzas Armadas ESPE, Av. General Rumiñahui s/n, Sangolquí 171-5-231B, Ecuador; mjerazo@espe.edu.ec; 2Arrhythmia Unit and Electrophysiology, Department of Cardiology, Virgen de la Arrixaca University Hospital, Instituto Murciano de Investigación Biosanitaria, 30120 Murcia, Spain; juanjosanchezmunoz@me.com (J.J.S.-M.); arcadi@secardiologia.es (A.G.-A.); 3Departamento de Teoría de la Señal y Comunicaciones, Sistemas Telemáticos y Computación, Universidad Rey Juan Carlos, 28943 Fuenlabrada, Spain; alicia.guerrero@urjc.es (A.G.-C.); estrella.everss@urjc.es (E.E.); joseluis.rojo@urjc.es (J.L.R.-Á.); 4Arrhythmia Unit, Department of Cardiology, Hospital General de Valencia, 46014 Valencia, Spain; quesada_aur@gva.es; 5Arrhythmia Unit, Department of Cardiology, Hospital Clínico Universitario, Av. Blasco Ibañez, 17, 46010 Valencia, Spain; ricardo.ruizgranell@gmail.com; 6Arrhythmia Unit, Department of Cardiology, Hospital Clínico de Alicante, 03010 Alicante, Spain; aliciacriado@gmail.com; 7Arrhythmia Unit, Hospital General de Castellón, 12004 Castellón, Spain; navarroa@medynet.com

**Keywords:** ventricular fibrillation, implantable cardioverter defibrillator, Rotor Theory, spectral analysis, entropy, mutual information

## Abstract

Ventricular fibrillation (VF) signals are characterized by highly volatile and erratic electrical impulses, the analysis of which is difficult given the complex behavior of the heart rhythms in the left (LV) and right ventricles (RV), as sometimes shown in intracardiac recorded Electrograms (EGM). However, there are few studies that analyze VF in humans according to the simultaneous behavior of heart signals in the two ventricles. The objective of this work was to perform a spectral and a non-linear analysis of the recordings of 22 patients with Congestive Heart Failure (CHF) and clinical indication for a cardiac resynchronization device, simultaneously obtained in LV and RV during induced VF in patients with a Biventricular Implantable Cardioverter Defibrillator (BICD) Contak Renewal IV^TM^ (Boston Sci.). The Fourier Transform was used to identify the spectral content of the first six seconds of signals recorded in the RV and LV simultaneously. In addition, measurements that were based on Information Theory were scrutinized, including Entropy and Mutual Information. The results showed that in most patients the spectral envelopes of the EGM sources of RV and LV were complex, different, and with several frequency peaks. In addition, the Dominant Frequency (DF) in the LV was higher than in the RV, while the Organization Index (OI) had the opposite trend. The entropy measurements were more regular in the RV than in the LV, thus supporting the spectral findings. We can conclude that basic stochastic processing techniques should be scrutinized with caution and from basic to elaborated techniques, but they can provide us with useful information on the biosignals from both ventricles during VF.

## 1. Introduction

Cardiovascular diseases are currently the leading cause of death worldwide. The World Health Organization (WHO) estimates that, in 2012, approximately 17.5 million people passed due to these diseases, and it is estimated that by 2030 almost 23.6 million people will pass from some cardiovascular disease, with the coronary heart disease (also known as myocardial infarction) being the most common cause [1]. The official data from the National Statistics Institute of Spain (INE) reported that in 2017 there were approximately 90,000 deaths due to heart problems, from which 15% corresponded to myocardial infarction. [2]. Ventricular fibrillation (VF) is a heart rhythm disturbance that occurs when the heart beats with very fast and erratic electrical impulses, causing the ventricles to agitate with ineffective pulsations instead of pumping blood, which may cause death unless reversed in the first few minutes [3].

The search of successful therapies for the prevention of sudden cardiac death has motivated the development of studies focused on the ventricular signal analysis during VF, in order to better understand the mechanisms underlying its apparently chaotic behavior. During the second half of the 20th century, the predominant hypothesis to understand the VF mechanisms was based on the random and disorganized propagation of multiple waves [4,5]. However, an alternative theory was proposed in 1994, which described a possibly three-dimensional electric rotor as the main organizing source of the cardiac functional re-entry activity in the VF context [6,7,8]. The available literature suggests that the electric rotors and their propagation in the form of spiral waves are either near-periodic, or non-linear, or both, in essence, so that the analysis of the electric rotor behavior could be addressed with tools derived from Spectral Theory and from Non-linear Systems [8,9,10]. Nowadays, the Rotor Theory proposes new challenges in terms of the development of new analytical techniques to quantify the VF dynamics, as well as the study of the temporal and spatial VF evolution. In this context, the literature suggests that the application of spectral analysis and non-linear processing techniques have contributed significantly to the detailed study of the properties of the electrophysiological signals during VF [11,12,13]. However, existing studies have been mostly conducted with limited data on intracardiac signals in humans with relevant clinical pathologies, where the VF could be generated through mechanisms very different from those of small hearts in experimental (animal) models. Whereas, the rotor hypotheses studied have been based based on experimental data only. Additionally, and to our best knowledge, there are few comparative studies on the electrophysiological characteristics of the electrical activities of Right Ventricle (RV) and Left Ventricle (LV) [14,15].

Currently, VF is medically treated with resynchronization devices, known as implantable cardioverter defibrillators. These devices are suitable for research purposes, as they allow the simultaneous and accurate recording of the electrograms (EGMs) of RV and LV during a VF event [16]. Based on this technologically available possibility, the objectives of this research were: First, to simultaneously analyze the intracardiac electrical signal of RV and the epicardial signal of LV during induced VF in the procedure of implantation of resynchronization devices with defibrillation capability; and second, to evaluate the similarities and differences between both ventricles in terms of basic spectral and nonlinear parameters. For these purposes, we used basic representations of the frequency spectrum, as well as fundamental calculations of the Shannon entropy and (Mutual Information) MI in said signals. Therefore, this study examines the possibility that the RV activation varies with respect to the LV activation during VF episodes, as well as whether the fundamental frequency is superior in LV, which would suggest a rotor anchored model in some LV structure with a mostly passive RV. The study of the local organization of the activation form in each ventricle represents a piece of information that could also support a rotor mechanism compared to a multiple reentry mechanism. In our study, 22 patients were included with Congestive Heart Failure (CHF) and with clinical indication for a cardiac resynchronization device with defibrillation capability, specifically the Contak Renewal IV^TM^ (Boston).

The intracardiac EGM signals of both ventricles were compared in terms of their spectral content and their shared information and, for this purpose, we used basic representations of the frequency spectrum, as well as the fundamental calculations of the Shannon entropy and MI in said signals. The contributions of the paper are twofold. First, the knowledge of fundamental information about the VF mechanism is scrutinized in human VF, which represents a scenario where much is still unknown. Second, the consistency of spectral and Information-Theory based measurements is also analyzed, which has been rarely addressed in the VF analysis literature.

This document is organized as follows. Section 2 summarizes the work related to the study of electrophysiological signals by spectral analysis and MI. Section 3 describes the methods of acquisition and generation of the patient database, and it explains the spectral and statistical analysis tools applied to identify potential differences between RV and LV data. Section 4 summarizes the results of the analysis of spectral content and the degree of association between the signals acquired for each ventricle and among patients. Finally, Section 5 presents the discussion and conclusions.

## 2. Related Work

The study of human electrophysiological signals has been the subject of special research interest. The intracardiac signals are not an exception, since one of the frequent causes of death in the world is heart diseases, such as the arrhythmias caused by VF events. For this reason, there are several studies that are focused on finding successful therapies for the prevention of sudden cardiac death. In this section, different research works are addressed, which were oriented to the application of spectral content study techniques as well as to non-linear analysis of electrophysiological signals.

The spectral analysis of cardiac signals has made possible to identify, in detail, the content of the harmonics of the cardiac rhythm when present, and from this, the periodic components included in the signal as peaks of the power spectrum have been quantified [17]. In this field, the Fast Fourier Transform (FFT) is one of the most used nonparametric techniques in the estimation of the Spectral Power Density (SPD) contained in the EGM signals, and it has allowed for an appropriate analysis of the significant frequency variations that occur in cardiac bio-electric recordings before state transitions, for example, between wakefulness and sleep [18,19]. The application of parametric statistics has also been one of the widely used methods in the spectral analysis of EGM signals, so that autoregressive models have been applied to find relevant information from these signals by creating algorithms that have detected, for example, premature ventricular contractions by means of artificial intelligence. In addition, autoregressive models develop studies where they could model non-stationary signals recorded during short periods of time, as is the case with intracardiac EGM during arrhythmic episodes [18,20,21].

Nonlinear analysis methods are being increasingly used in the study of cardiac biopotentials, especially in cases, such as arrhythmias, whose signals are characterized by nonlinear and nonoscillatory spectral components, conditions that can be similar to those of some nonlinear dynamic systems [22,23]. Several works have been developed describing non-linear analysis techniques for physiological signals, whose evaluation is based on developing methods inspired by Chaos Theory and Non-linear System Dynamics, such as Fractal Dimension (FD), Correlation Dimension (D2), or the Largest Lyapunov Exponent (LLE). Other indices have been proposed based on the application of Information Theory principles oriented to the use of metrics, such as Renyi’s Entropy (REN), Shannon’s Spectral Entropy (SEN), and Approximate Entropy (ApEn) [12]. The entropy is considered as the basis for one of the main methods of nonlinear analysis, since it quantifies the degree of uncertainty of a physiological signal and, for this reason, it has been previously used in the analysis of VF signals [24].

The above presented background shows that the research developed on the spectral and nonlinear analysis of VF waves is comprehensive, and it has improved our knowledge on aspects, such as the initiation mechanisms [11], the acute response to various drugs [25,26], the predictive factors for successful defibrillation [27], and the effects of the infarcted area location [28]. However, these studies have been mostly conducted for the RV [29] and, to our best knowledge, there are few comparative studies on the electrophysiological characteristics of the electrical activities between RV and LV. In this context, the MI can represent a valuable tool for the cross-information analysis of the VF signals in both ventricles, since it allows for the quantification of the dependency degree or the information amount that can be obtained on a random variable through another one [30]. Nevertheless, MI studies that involve electrophysiological signals are scarce, and the majority of them are oriented to the analysis of the association between the decrease in respiration with respect to myocardial infarction [31], as well as to the comparison of images between ventricles [32]. Other experimental models have limited themselves to the description of differences, such as a greater degree of organization in the RV [33] and a greater number of reentry waves in the LV [34].

Currently, VF is treated medically with resynchronization devices known as internal cardiac defibrillators. The function of this device is to place electrodes in the internal heart chambers which capture several bioelectric signal. These defibrillators are placed in patients with CHF, and in addition to their resynchronization role, they are also capable of detecting cardiac arrhythmias and recognizing when VF is involved, while accurately recording the EGM of RV and LV during VF in humans [35].

## 3. Methods

### 3.1. Intracardiac Electrogram (EGM) Signals and Ventricular Fibrillation (VF)

Heartbeats are caused by electrical impulses as a result of exchanging ions such as sodium, potassium, or calcium, among others. The electrical impulses propagate through the heart tissues, thereby generating small currents on the cardiac cells and fibers [36]. This behavior allows for specialized devices as the EGM to record the bioelectric potential variations with time and to represent the corresponding cardiac rhythm waveforms. A typical EGM signal from a healthy heart is characterized by successive waveforms, known as *P wave, QRS complex and T wave,* as illustrated in Figure 1 (left). The signal portion between the *P* wave and the *QRS* complex is called the depolarization wave, because the *P* section represents the electrical potentials that appear when the atria are depolarized before their contraction, while the *QRS* complex represents the electrical potentials that are generated when the ventricles are depolarized before their contraction. The *T* wave section, on the other hand, is known as the repolarization wave, since it represents the electrical potentials generated when the ventricles recover from the depolarization state, also known as repolarization [37]. The morphology of the *P-QRS-T* waves can be modified by the presence of some serious cardiac pathology, such as VF, which can be defined as a spontaneous and asynchronous and local contraction of cardiac muscle fibers that prevents the effective and global ventricular contraction. This causes blood pressure to drop rapidly, which interrupts the blood supply to vital organs. Figure 1 (right) shows a typical EGM waveform during VF, where the absence of organized ventricular depolarization modifies the *P-QRS-T* pattern, which is completely absent. In these circumstances, the heart rate is extremely irregular, producing uncoordinated electrical activity of the heart that can cause death within seconds [3].

The VF is the main cause of sudden cardiac death and, unlike of other arrhythmias, it is considered to be pharmacologically unapproachable, since it seems a succession of uncoordinated and chaotic electric fronts. During several decades, the VF prevention strategies have principally focused on the suppression of the ectopic ventricular beats, which could even precipitate the disease. However, the clinical results reported in the literature have not contributed significantly to the reduction of this cardiac condition [38,39,40,41]. Nowadays, the global vision of VF and Atrial Fibrillation (AF) has changed dramatically according to the so-called Rotor Theory concepts [42,43,44]. This theory states that the VF is derived from a phenomenon known as *reentry*, which would be given here by the helical and unbroken circulation of electrical impulses, either anatomical or functional. These impulses would rotate following dynamics determined by an organizational pivot, they are commonly called *rotors*, and they generate waves with excessively high component frequencies as compared with the sinusal rhythm, so that ineffective cardiac contractions are provoked. The rotors behave as rotation centers that produce multiple electrical activation fronts at some distance from them, and they are conditioned by the myocardial electrophysiological properties, which, in turn, depend on the different ionic current dynamics. Early research revealed that rotors play a significant role as VF producing agents in both animal and human models [8,45,46]. However, there are few studies in patients where VF signals are measured in both ventricles simultaneously.

### 3.2. VF Signals Detection and Patient Database

This study was performed on patients with CHF problems using the same equipment to take data from the detection of VF. The equipment used to measure the signals from the ventricles is known as Cardiac Resynchronization Therapy (CRT) device; it senses the heart signals from the two ventricles and, when necessary, it delivers an electrical shock to the heart to restore the extremely fast and sometimes irregular heart rhythm to a normal rhythm. CRT devices convey a small computer powered by an internal battery, securely sealed inside the box. The device continuously monitors the heart rhythm by recording information and it provides electrical therapy to the heart when it detects an arrhythmia. Several device types have been developed, including pacemakers (CRT-P) and defibrillators (CRT-D). In this research, we used the CRT-D device model Contak Renewal IV^TM^ from Boston Cientific Corporation. The CRT uses isolated electrocatheters connected to the device and implanted in the heart. The wire transmits the signal from the heart to the device, and then it carries the energy from the device to the heart to coordinate its rhythm [47].

The device has five electrical connectors arranged, as follows: two defibrillation output connectors with DF-1 connector generate electrical shocks that are located in the RV; three sensor input connectors are connected to the RV with IS-1 connector, to the LV with LV-1 connector, and the third is connected to the Right Atrium (RA) with an IS-1 connector. The medical procedure applied by professionals in the field indicates that the sensor cable connection goes to the LV, the defibrillation and sensor cables are connected to the RV, and the sensor cable going to the RA remains disconnected at all times [48,49], as shown in Figure 2.

For data acquisition, two types of electrodes were used with Contak Renewal IV, which were the Easytrak 2 model (proximal and distal electrode areas, 4.2 mm${}^{2}$ and 4 mm${}^{2}$, respectively), and the Easytrak 3 model (proximal and distal electrode areas, 9 mm${}^{2}$ and 8.5 mm${}^{2}$, respectively). Both of the models had an electrode spacing of 11 mm. Several probe models were implanted in the RV, all with the same characteristics compared to the stimulation/detection electrodes (distal electrode area 2 mm${}^{2}$, proximal defibrillation coil electrode 450 mm${}^{2}$, electrode spacing 12 mm). VF was induced with a T-wave shock of 1.1 J in all patients, and all the other induction parameters were set at the discretion of the attending physicians. The gain of the device amplifier was optimized to avoid very small amplitude recordings or saturation. The ECG signals were simultaneously recorded on a continuous strip of paper at 50 mm/s, as shown in Figure 3.

The data that were obtained on the continuous paper were digitized through a customized software in order to erase the grid and export the RV and LV signals in ASCII format. We denote the RV signal as $x_{R}\left(t\right)$ and the LV signal as $x_{L}\left(t\right)$ [50]. The sampling period of the restored EGM was 5 ms (sampling frequency of 200 Hz).

During the implant procedure in patients, electric shocks were delivered to the ventricles generating a controlled VF episode in order to test the device functionality, and these signals were recoreded through the sensors. Two tests were performed in most patients, but only the first recording was included in the study to avoid possible differences due to consecutive shocks. We also recorded demographic data (age and sex), heart disease (ischemic, dilated cardiomyopathy, others), drugs at the time of implantation, echocardiographic parameters, and implantation data (including minimum energy and number of shocks for successful defibrillation).

### 3.3. Spectral Measurements

Biomedical signals in time are the result of biopotentials changing in the spatial and in the temporal domains. However, in some cases it is desirable to study them in the frequency domain, both in deterministic and in stochastic problem statements. Frequency analysis can be performed while using representations based on the Fourier Transform of a given signal, for instance, for the LV recorded signal in a patient, we have that
(1)$$X_{L}\left(f\right)=\int_{-\infty }^{+\infty }x_{L}\left(t\right)e^{-j2\pi ft}dt$$
where $X_{L}\left(f\right)$ is the LV signal in the frequency domain. Spectral analysis is a mathematical and statistical tool that uses the Fourier Transform as its basis, and it allows us to estimate the Power Spectral Density (PSD) of a signal, which represents the average power distribution of the signal in the frequency domain, and it is calculated as $P_{L}\left(f\right)={\left|X_{L}\left(f\right)\right|}^{2}$ for the example of the LV signal.

From the Fourier Transform, we can measure parameters that have been previously proposed in the literature, such as the Fundamental Frequency (FF) [51], the Dominant Frequency (DF), the Harmonic Frequencies (HF), denoted by $f_{0}$, $f_{1}$, $f_{k}$(k=2,3,4...), respectively, and we also can determine their corresponding Spectral Peak (SP) amplitudes [51], as shown in Figure 4. Besides, we obtained the Mean Frequency (MF), denoted as $f_{mean}$, defined as the gravity center of the spectrum, and being closely related to the profile of the spectral envelope, especially in cases where a harmonic structure is present. The Bandwidth (BW) was calculated according to its definition as the difference between the upper and lower frequencies for which the maximum spectral power remained above its 75% [52]. The Organization Index (OI) has been defined in the VF literature (from the previously proposed measurements of spectral regularity) as the ratio of the sum of the power in the harmonic bandwidths in the 2–30 Hz band to the total power in this band [53,54]. The Leakage (LK) has been defined as the correlation coefficient between the EGM registered and a sinusoidal function with the same fundamental period and phase adjusted to maximize this coefficient, so that this parameter measures the shape similarity between a EGM signal and a sinusoidal signal [55].

For our research, we used the periodogram proposed by Welch, which consists of dividing the time signal into overlapping segments at 50%, finding the Fourier Transform for each segment, and averaging the estimated windowed spectral densities. The averaging in the periodgram tends to reduce the estimator variance over the entire time series. On the other hand, although the overlap between consecutive segments introduces redundant information, this effect can be minimized by the use of non-rectangular windows that reduce the importance or weight given to the final samples of the segment [56]. Spectral Analysis of the EGM was made using the averaged Welch periodogram with 256-sample rectangular window for maximum spectral resolution, 4096 samples for the Fourier representation support, and 50 % overlapping between successive windows. In order to compare EGM signals from different patients and with possibly different amplitudes, each periodogram $P\left(f\right)$ was normalized to unit area, denoted as power spectrum $P_{n}\left(f\right)$. The EGM characteristics were analyzed for the RV and LV, during the first three seconds and in the interval between three and six seconds.

### 3.4. Information Theory Measurements

Information Theory is the mathematical treatment of the concepts, parameters, and rules governing the transmission of messages through communication systems. It was founded by Claude Shannon toward the middle of the twentieth century and has since then evolved into a vigorous branch of mathematics, while fostering the development of other scientific fields, such as statistics, biology, behavioral science, neuroscience, or statistical mechanics. The techniques used in Information Theory are probabilistic in nature, so that some authors see Information Theory as a branch of probability theory [43,57]. The development of the entropy concept for random variables and processes by Shannon provided the beginning of Information Theory and of the modern age of Ergodic Theory. The entropy and its related information measures provide useful descriptions of the long term behavior of random processes, and this behavior is a key factor in developing the coding theorems of current digital communication systems. We introduce next the notions of entropy for random variables and we summarize some of its fundamental properties [58].

The entropy is conceptualized in general terms as a natural trend measure of the order loss or peculiarity of certain combinations, and it is a widely used concept in fields, such as thermodynamics, statistical mechanics, and astrophysics, among others. In the Information Theory field, the entropy (also called information entropy or Shannon entropy) is a measure of the uncertainty degree that exists on a data set generated from a random experiment, as well as the necessary information to limit, reduce, or eliminate this uncertainty in any stochastic process. The Shannon entropy is also defined as the average rate at which the information is produced by a stochastic data source, and for a stochastic process (e.g., $x_{L}\left(t\right)$) it can be expressed as
(2)$$H\left(x_{L}\right)=-\int_{-\infty }^{\infty }p\left(x_{L}\right)log_{2}\left(p\left(x_{L}\right)\right)dx_{L}$$
where $H\left(x_{L}\right)$ is the entropy of the LV stochastic process, and $p\left(x_{L}\right)$ is the probability density of the LV stochastic process.

The Joint Entropy measures how much entropy is contained in a joint system of two random variables or, alternatively, it is the entropy of a joint probability distribution for a multivalued random variable, as given by
(3)$$H\left(x_{L},x_{R}\right)=-\int_{-\infty }^{\infty }\int_{-\infty }^{\infty }p\left(x_{L},x_{R}\right)log_{2}\left(p\left(x_{L},x_{R}\right)\right)dx_{L}dx_{R}$$
where $H\left(x_{L},x_{R}\right)$ is the joint entropy of cardiac signals from the ventricles, and $p\left(x_{L},x_{R}\right)$ is the joint probability of cardiac signals from the ventricles for each process.

The MI of two random variables is a measure of the mutual dependence between those two variables. More specifically, it quantifies the amount of information obtained by approximating one random variable through the observation of the other random variable. The concept of MI is herein measured as related to the entropy and the Joint Entropy of the cardiac signals, and it can be obtained as
(4)$$I\left(x_{L},x_{R}\right)=H\left(x_{L}\right)+H\left(x_{R}\right)-H\left(x_{L},x_{R}\right)$$
where $I\left(x_{L},x_{R}\right)$ denotes the MI between both cardiac signals. The MI has been used in a variety of practical applications, for example, to assess how environmental variables independently influence the interaction between ecosystem services, or to investigate the relation between earth surface temperatures and the spatial pattern of green space effects. In our research, the MI is applied to the analysis of similar interaction between cardiac signals that are acquired in the CRT-D device [59].

Note that, in this work, we did not use simplified and high-dimensional states estimators of information theory measurements, such as sample entropy, approximate entropy, or multiscale entropy. These measurements often require an extremely large number of time samples for their calculation, and it is clear that the VF signals are essentially short. Instead, we wanted to start from the very simplest way of estimating the Shannon Entropy, which corresponds to the previous equations. One should take into account that we are dealing with continuous random processes, which means that we should use the Differential Entropy, i.e., based on the direct discretization of said equations. Synthetic examples made evident that bias would be present when analyzing our data, due to the reduced amount of samples. For estimating the Differential Entropy (Shannon entropy on a continuous-distributed random process), several practical considerations need to be taken into account. The algorithmic implementation can be obtained by any interested reader under mail request.

## 4. Results

### 4.1. Spectral Measurements on Patients

Twenty-two patients (63.2 ± 11.5 years old, 15 men) had dilated cardiomyopathy (four ischemic, 18 idiopathic) in NYHA II (*n* = 3) or NYHA III (*n* = 15) status. Four patients were on amiodarone treatment.

Table 1 shows the averaged and standard deviation values for the analyzed spectral parameters, where paired Student t-tests were used to obtain the *p*-values. The spectrum of VF was complex, and several frequency peaks could be observed in most patients, showing a DF higher than FF. Characteristics of the right and left ventricular recordings were analyzed within the first three seconds of the episode and at the three to six seconds intervals, as summarized in the table. Differences between DF and MF, as well as differences among the power of harmonic peaks ($P_{n}\left(f_{2}\right)$, $P_{n}\left(f_{3}\right)$, $P_{n}\left(f_{4}\right)$), could be attributed to the effect of different spectral envelopes for each lead configuration.

Figure 5a shows an example of RV and LV simultaneously recorded EGMs, together with their normalized spectra, for a representative case. Figure 5b depicts the averaged spectrum for all the cases of VF (first three seconds), by using two different spectral estimators. In both, the averaged spectrum is shown to be more low-pass in the RV and more high-pass in the LV. Though, this could be attributed to differences in lead configuration, as far as the LV configuration (tip-ring) usually has a more marked low-pass character than the RV configuration (can-coil), which makes the differences on the underlying arrhythmic mechanism difficult to uncouple of the lead configuration. This difference in the lead configuration can also affect the differences between the OI in the LV and in the RV, as far as can-coil configuration has been shown to have larger OI than the same VF in the tip-ring lead recording. The DF was significantly higher in the LV during both analyzed periods, but FF did not show significant differences. The spectral peaks and the organization indices (OI and leakage) were higher in the RV recordings. The MF was higher in the LV. The values and statistical significance levels did not change noticeably when repeating the analysis by excluding the patients with amiodarone.

It is very likely that some of these results can be dependent on intrapatient characteristics related with structural and bioelectrical properties of the heart, which cannot be available unless very detailed experimental protocols (including medical image or intracardiac electrophysiological scans) are performed, which would modify the clinical management. It also can be seen (not shown) that spectra among different patients can noticeably differ. However, the consistency of the spectral parameters points out to the underlying bioelectrical processes being not too complex in some sense from the spectral point of view. The basic periodicity structure is in general well captured by the set of a fundamental frequency, its harmonics, and the spectral envelope (which of course are closely related by Spectral Analysis Theory), whereas the fluctuations from this periodicity are captured by the bandwidth, the organization, and the LK parameters. The significance of parameters mostly holds from the first three seconds to the next three seconds. Whereas, the spectral envelope has relevant differences, the DF would represent an inconsistent parameter, as it depends on the envelope (it could correspond to different harmonics in different patients), but the fundamental frequency is the basis for the near-to-periodic structure of the signals that can be recognized. Note that the configuration of sensing electrodes could be a confounding factor on the spectral envelope, but parameters as FF and organization measurements should be less dependent from spectral envelope signatures due to different electrode configurations.

### 4.2. Information Theory Measurements on Simulations

The parameters calculated from Information Theory principles are usually obtained by discretizing the equations presented in Section 3.4, as far as we are working with the continuous distributions. This means that we are using the differential entropy for continuous random variables, and that the accuracy of the measurement parameters will be dependent on at least two discretizing elements, namely, the bin distance for discretizing the density functions, denoted as Δd, and the number of available samples, denoted as *N*. We found that the choice of these parameters in the literature often is presented as heuristically tuned, whereas their choice can have a definitive impact on the final accuracy and quality of the parameter estimations.

We performed a set of simple experiments to previously check that the free parameter setting was adequate in order to estimate the entropy of each of two processes, their joint entropy, and their MI. For this purpose, we generated a bivariate Gaussian process with zero mean and covariance matrix, given by
$$\left(\begin{array}{cc} 3 & 1\\ 1 & 0.5 \end{array}\right)$$

A different run of values was generated for each value of *N*, between $N=10^{2}$ and $N=10^{6}$ in logarithmic scale. For each run, the bin size for the estimations was swept between $\Delta b=10^{-3}$ and $\Delta b=10^{0}$ in logarithmic scale. Figure 6 shows the results for a simple experiment in which signals are totally dependent. It can be seen that, for *N* large enough, we obtain good quality estimation for alll or the parameters and for almost any bin size. For a moderate number of available samples, it is still possible to obtain good quality estimators as far as the bin size is not too small, whereas for a low number of available samples there will be an unavoidable and uncontrolled bias on the estimators.

In a real case, we will have to work with a fixed number of samples *N*, but we can still scrutinize the effect of changing the bin size on the estimated parameters and scrutinize whether said estimation reaches a stable point, so that we are either in the middle region of this kind of graph or we are in a biased region. This information will be useful to determine the adequate bin size on these estimators.

### 4.3. Information Theory Measurements on Cardiac Signals

A second experiment was performed to measure the MI in the analysis of cardiac signals to scrutinize the relationship between recordings in each ventricle for each patient. The number of samples *N* for each signal is previously fixed in real recordings, so we used all of the available samples for each signal, and we swept the bin width Δb similarly to the previous experiment. Figure 7 shows the measurements for each patient. It can be seen that entropy measurements are more regular in the LV signals than in the RV signals; nevertheless, none of them reaches a plane region for any bin sizes, which indicates that we have few available samples and bias will be present in the measurements. Nevertheless, the Joint Entropy shows that it is close to its stabilization for larger bin sizes, whereas the MI shows more variance for those regions. This can be seen as an indicator that LV signals, in general, are more variable among patients than RV signals, and also that the Joint Entropy and the MI have different estimation properties in terms of sensitivity to variance.

According to the behavior observed in the Information-Theory based measurements on the synthetic data experiments, we should chose a bin width as large as possible without introducing distortion, and given the strong presence of bias due to the moderate number of available time samples, we choose the same bin width for all the signals, aiming to control this bias for this population. We checked several bin widths (0.6952, 0.4833, 0.3360, and 0.2336), and the populational behavior was similar in them. As it can be seen in Figure 8, larger bin widths lead to higher MI estimates in the population (most of them above unity), but they are mostly over-smoothed; hence, the measurement looses its sensitivity among patients. With more reduced values of the example bin widths, there is an increase in sensitivity, in the sense that a larger range is covered in the population, but, in several cases, the MI goes to zero or very close, which could be mostly attributed to increased variance of the estimator. The intermediate value 0.4833 seems to provide an acceptable trade-off between bias and variance, with sensitivity to account for the differences among patients in the population, and consistent with the peaks and valleys in the larger and smaller bin widths.

Figure 9 shows the populational results when choosing Δb=0.4833 as free parameter tuned for the estimations in all the measurements in all the patients. We can confirm that the Entropy in the LV is more variable than in the RV and, in general, it is at most as high as the Entropy in the RV, as their mean ± standard deviation were 1.408±0.295 in the LV vs 1.5719±0.1968 in the RV (p=0.009 with paired *t*-test). This seems to be consistent with the existence of more complex bioelectric activation in the LV as compared with the RV, although the differences in the configuration of the electrodes is again a possible confounding factor that should be taken into account. Note that the Joint Entropy is conformed to be a slightly smoothed version of the combination of both entropies, but not just their summation, and that the MI is, in general, weakly correlated with any of the previous components. This indicates that Entropy, Joint Entropy, and MI are measuring different aspects or the information content in these signals.

## 5. Discussion

In this work, we have presented a spectral and a non-linear analysis of the LV and RV signals simultaneously acquired during induced VF in 22 patients with CHF and clinical indication for a BICD. During the induced VF in the implant procedure, the EGM obtained between the tip and ring of the RV electrodes and the EGM of the bipolar epicardial electrode in the LV were recorded. The Fourier representation of the first six seconds of the signals recorded for RV and LV made it possible to identify spectral parameters, whereas the Information Theory content both in individual and grouped recorded signals was scrutinized using Shannon entropy and MI. Our analysis showed that VF spectra were complex and several frequency peaks were observed in most patients, showing a DF that was greater than the LV than in the RV, while the OI had the opposite trend. With respect to the entropy and MI study, we concluded that, in all cases, the reduction in the bin size of the histogram improved the association between the RV and LV signals, as evidenced by the progressive increase in Entropy and MI until maintained value approximately constant and independent of that continue reduction of the histogram bin size. When accounting for these aspects on the estimators in Information Theory parameters, the LV was checked to be more variable in terms of Entropy, whereas the RV was checked to be smoother, on a trend which was consistent for all the patients.

After early interest of studies in VF on advanced Information-Theory based measurements, they have not received so much later attention as the spectral-based measurements. In this work we have scrutinized basic entropy-related measurements, and they have proved to be informative in this scenario. However, a word of caution should be kept, as far as the simple consistency check performed in our work is often lost in the literature. We have seen that the number of samples is bringing variance and bias in the real signals, which is in accordance with the sensitivity analysis on synthetic signals for the number of samples and the bin size. Other methods could be used to improve the consistency of these kind of indices, for instance, the use of Parzen window estimators for deriving the entropy could be reducing bias and variance. Therefore, it seems recommendable to use several types of features, including temporal, spectral, and entropy-based characteristics for VF signals, in order to give a wider view of the underlying dynamics.

The measurements obtained herein showed significant differences in the spectral characteristics of the electrical signal from the RV and LV as recorded by the electrodes of a biventricular resynchronization device during induced VF. Specifically, the DF was higher in LV, whereas the organization was higher in the RV. A different pattern of the ventricular activation signal in RV and LV has been reported in experimental settings. In a swine model of induced VF with epicardial recordings, Rogers et al. [33] found less wavefronts and a slower activation rate on the RV and, also, VF patterns in the RV were significantly more organized than those in the LV. Our results were partially in agreement with these experimental findings, since lower DF and higher organization were found in the RV recordings, but we did not find differences in the FF, which better characterizes the averaged periodicity of the analyzed EGM. The geometric properties of the left and right ventricles could play a role in the different behavior of the VF signal. It has been hypothesized that reentry in thin tissue layers tends to be stable, whereas, in thick preparations, can more easily degenerate in VF [6]. In the LV wall, the core filaments of the three-dimensional rotors would have place to deform and split giving rise to multiple wavefronts, fragmentation, and more irregular electrical activity [33], which also could explain the lower degree of organization in the LV signals. Other anatomic structures of the LV have been related to the initiation or maintenance of VF. In an experimental study in dogs, the highest DF during VF was located on the posterior papillary muscle, and radiofrequency ablation at this area highly reduced the inducibility of VF [60]. The potential role of the posterior papillary muscle as an anchoring point for the rotors maintaining the VF could be another reason for a higher DF in the LV. The epicardial and endocardial locations of the left and right electrodes, respectively, can also play a major role in the observed differences. In contrast to our results, a trend to-ward the presence of faster rotors on the endocardium compared with on the epicardium has been reported in human hearts [61]. This apparent disagreement could be explained by the assumption that the VF originates in the LV and the DF decreases gradually through the interventricular septum, as described in other models [54,62]. Regional differences in the genetic expression of several ionic currents may also induce a gradient of excitation frequencies from the LV to the RV during VF [54] and be responsible for the lower DF found in our series. The differences in DF and MF, as well as in the power of harmonic peaks, can also be attributed to the effect of different spectral envelopes for each lead configuration, since the LV configuration (tip-ring) usually has a more marked low-pass character than the RV configuration (can-coil), which makes the differences on the underlying arrhythmic mechanism difficult to uncouple of the lead configuration. This difference in the lead configuration can also affect the differences between the OI in the LV and in the RV, as far as can-coil configuration has been shown to have larger OI than the same VF in the tip-ring lead recording [42]. The study of VF characteristics using EGM in experimental models has also shown the significant effect of unipolar vs bipolar lead configuration [63,64,65] in this kind of recordings, in terms of the spectral parameters. In a previous work [66], the presence of frequency gradients from LV to RV was reported for bipolar recording configurations in resynchronization devices. These results cannot be compared to ours, given that the bipolar configurations in that study had different tip to ring separations, devices were from different companies, and the signal processing method was different. Specifically, EGM were preselected according to a quality criterion in terms of averaged periodicity and spectral parameters, whereas the case inclusion in our study has been fully independent of previous considerations about the quality of the spectral parameters. Other works on the EGM spectral parameters [67], in which the lead configurations were perfectly comparable, have allowed to obtain conclusive results, which should be taken into account for studies related to spectral characterization of EGM stored in implantable devices or recorded in catheters during VF.

The DF of VF bioelectric recordings has been shown to be higher in the LV than in the RV in some experimental models, thus suggesting that discrete, high frequency, and relatively organized sources of electrical activation in the LV may be responsible for the maintenance of the arrhythmia, being the RV a bystander. Very few data are available on the literature to know whether this hypothesis can be extrapolated to humans. Our study provides data from patients with VF and suggests that the arrhythmia mechanisms may be similar to those ones described in experimental models. Some consequences of this conclusion are that potential approaches to prevent VF using ablative techniques, as well as new techniques for effective defibrillation, should focus on the left ventricle. Furthermore, the informative parameters can be easily obtained in the available resynchronization devices, and their analysis might help in the distinction between VF and other irregular rhythms with high ventricular rate, such as atrial fibrillation, or even noise due to an electrode dysfunction, often being erroneously interpreted as VF by contemporary devices.

The present problem represents an scenario where acquiring, digitizing, and analyzing bioelectric signals can be useful for compiling basic knowledge on a relevant scenario (VF mechanism elucidation for subsequently elaborating new therapies for it). However, the sensors available in an informative group of patients (CRT devices) have been designed for another purpose (resynchronization sensing and therapy), which should be preserved. The point raised in the sensors arena is whether it still can be possible to obtain reliable and informative measurements with respect to the dynamics of this arrhythmia in each ventricle despite each ventricle has a different sensor signature for signal acquisition. In this case, the solution that we can reach comes from using a diversity of measurement principles (spectral analysis, Information Theory) and a diversity of parameters (near-periodicity and its fluctuations).

The main limitation of the present study is the limited availability of recordings in the LV and RV, which precludes a more accurate estimation of the activation characteristics in both chambers. Moreover, the signals had to be recorded with different electrodes and from epicardium in the LV versus the endocardium in the RV, hence introducing possible confounding factors in the results. These limitations are intrinsic to the recording procedure, and we aimed to compensate for it to some extent by using measurements of different nature, namely, spectral and Information-Theory based measurements, which exhibited consistent results between themselves. Although the nature of the EGM lead configurations is sometimes obviated in studies in implantable devices, our study indicates that the effect of lead configuration can have a highly relevant effect on the periodicity and organization indices when using spectral measurements. Finally, the results apply to episodes of VF that are induced from the RV apex, but might not extrapolate to the spontaneous episodes or to other induction methods. From the clinical point of view, the DF has been related to the defibrillation threshold, [68,69] and this work states the question about the existence of a different relationship between the DF of the RV or the LV with respect to that threshold. The spectral characteristics of the VF in the RV and LV in human beings are different and, although they might be explained by the existence of a fast source in the LV during VF, the sensing electrode configuration can be a blurring factor in these differences, which should be taken into account in future research. Nevertheless, CRT devices provide a way to get these signals, otherwise hard to obtain simultaneously in patients. The present work aimed to compensate this limitation by using different-based measurements with carefully designed preprocessing and feature extraction, which was helpful in this scenario.

We conclude that signal processing techniques based on spectral-analysis and entropy-based features in intracardiac recordings can be useful for analyzing the underlying dynamics of VF, which today and after years of intense research, remains unknown. Caution should be taken with Information-Theory measurements and the bias and variance properties of their estimators, but they can then provide useful information. Differences are obtained between the nature of the signals in the RV and in the LV, and though the sensing configuration could be a confusing factor, still these differences seem consistent across parameters of different kinds in stochastic processing. 

## Figures and Tables

**Figure 1 sensors-20-04162-f001:**
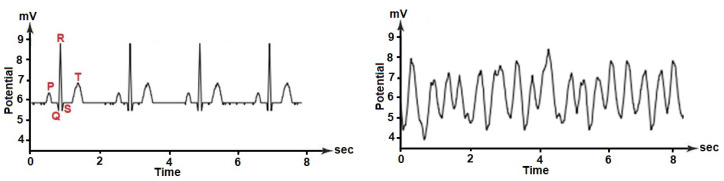
Example of ECG signals during normal heart rhythm (**left**) and during VF (**right**).

**Figure 2 sensors-20-04162-f002:**
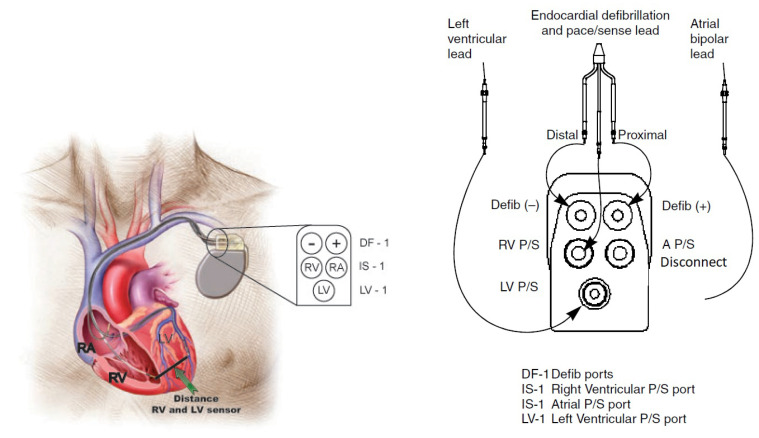
CRT-D device inserted into the ventricles of the heart (**left**) and connections in the device (**right**).

**Figure 3 sensors-20-04162-f003:**
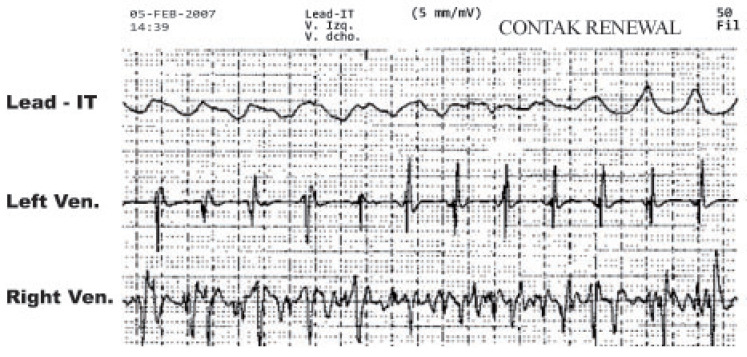
Example of recorded and stored EGM.

**Figure 4 sensors-20-04162-f004:**
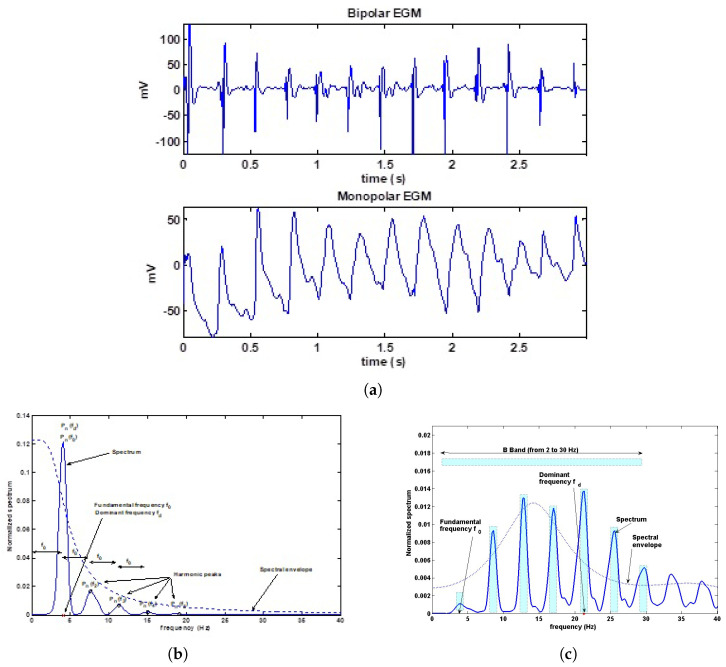
Example of EGM parameters in the frequency domain. (**a**) Monopolar and bipolar EGMs in the time domain. (**b**) Spectrum parameters in a typical monopolar recording. (**b**) Spectrum parameters in a typical bipolar recording. (**c**) The OI is obtained by the ratio between the power in the bandwidth of the harmonic peaks in the B band (2–30 Hz), and the total power in the band.

**Figure 5 sensors-20-04162-f005:**
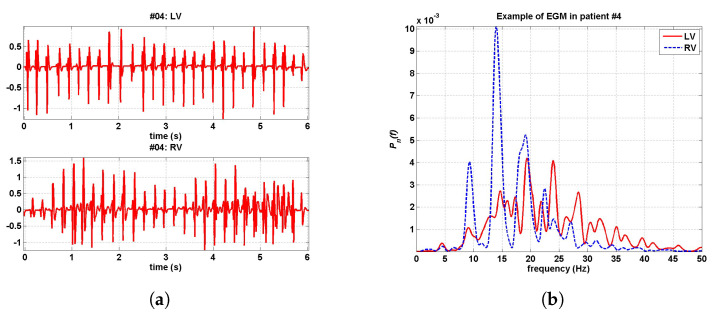
Example of EGM in RV and LV, in the time and frequency domains. (**a**) EGM simultaneously recorded in RV and LV. (**b**) Their corresponding spectra normalized to unit area.

**Figure 6 sensors-20-04162-f006:**
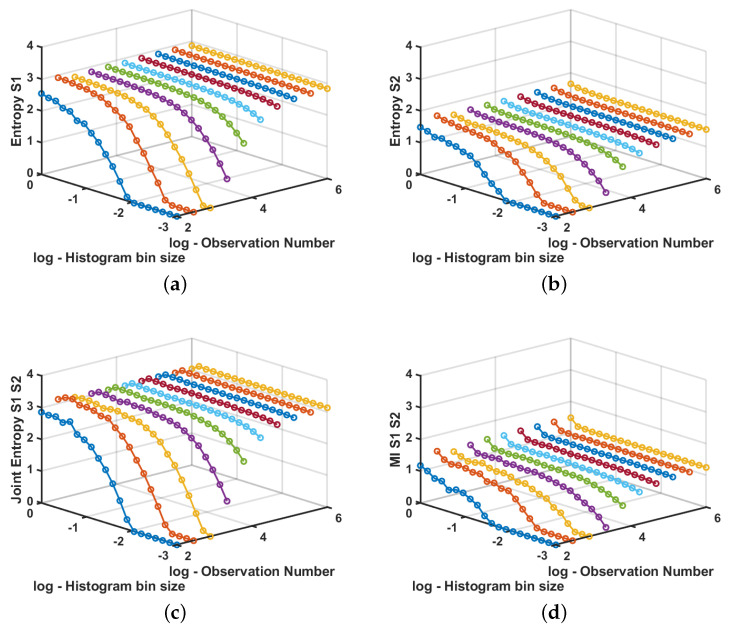
Example of entropy, joint entropy, and MI estimation in synthetic data signals. (**a**) Estimated entropy for the first random process. (**b**) Estimated entropy for the second random process. (**c**) Estimated joint entropy between both processes. (**d**) Estimated MI for the presented example.

**Figure 7 sensors-20-04162-f007:**
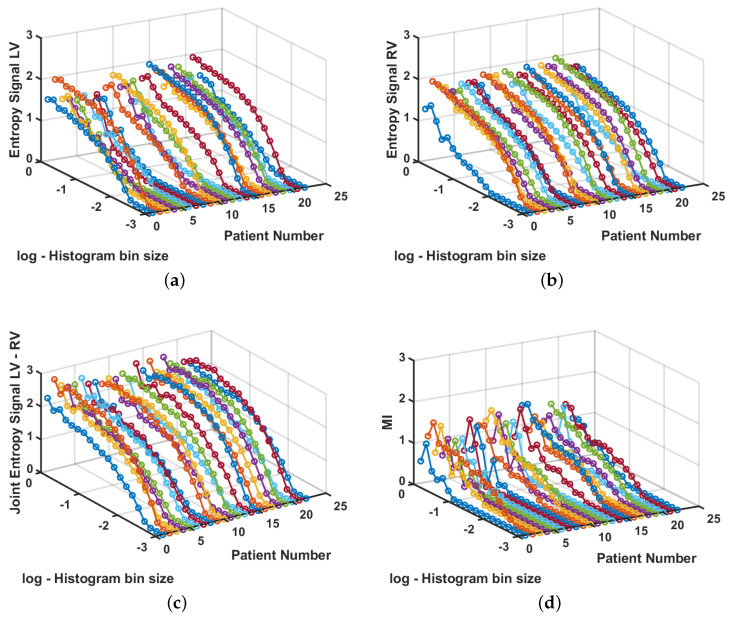
Entropy, Joint Entropy, and MI estimation in cardiac signals. (**a**) Entropy for signals from the LV. (**b**) Entropy for signals from the RV. (**c**) Joint entropy for signals from LV and from RV in each patient. (**d**) MI for signals from LV and RV.

**Figure 8 sensors-20-04162-f008:**
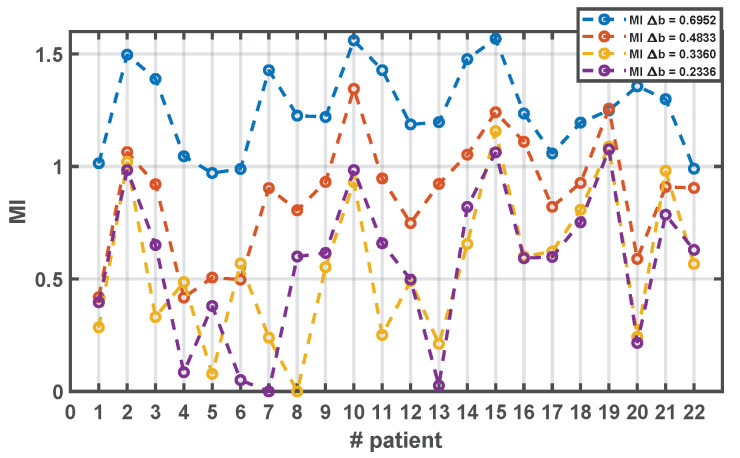
Estimated MI for cardiac signals in patient database for different bin widths.

**Figure 9 sensors-20-04162-f009:**
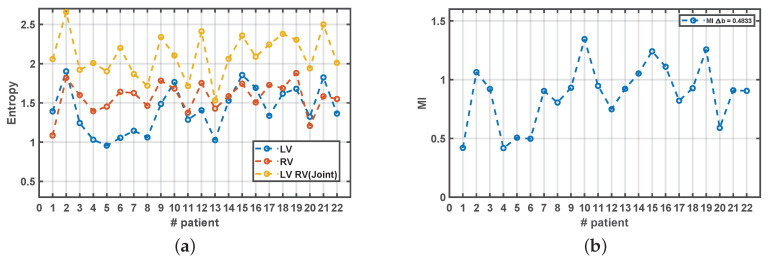
Measurements from Information Theory for cardiac signals in the patient database, using Δb=0.4833. (**a**) Estimated Entropy for LV and RV, and their Joint Entropy. (**b**) Estimated MI.

**Table 1 sensors-20-04162-t001:** Spectral parameters for the two analyzed time periods in the VF episodes.

Parameter	Time Period: 0–3 s			Time Period: 3–6 s		
	LV	RV	*p*	LV	RV	*p*
$f_{0}$ (Hz)	4.74 ± 0.62	4.74 ± 0.43	ns	4.83 ± 0.64	4.83 ± 0.47	ns
DF (HZ)	19.72 ± 4.90	15.69 ± 3.37	0.004	20.45 ± 4.79	15.66 ± 4.99	0.006
$f_{mean}$ (Hz)	22.45 ± 4.92	22.45 ± 3.70	<0.001	25.97 ± 4.01	22.25 ± 3.45	0.001
$P_{n}\left(f_{0}\right)$	0.54 ± 0.56	0.53 ± 0.66	ns	0.62 ± 0.50	0.57 ± 0.44	ns
$P_{n}\left(f_{1}\right)$	5.60 ± 2.62	8.27 ± 2.66	<0.001	6.06 ± 2.26	7.70 ± 2.69	0.022
$P_{n}\left(f_{2}\right)$	1.95 ± 2.09	4.10 ± 2.00	<0.001	2.14 ± 1.98	5.01 ± 3.31	0.002
$P_{n}\left(f_{3}\right)$	4.29 ± 3.11	7.67 ± 3.09	0.002	4.42±2.79	6.29 ± 3.05	0.02
$P_{n}\left(f_{4}\right)$	4.04 ± 1.61	5.78 ± 1.88	0.002	4.66 ± 1.90	5.08 ± 2.06	ns
$P_{n}\left(f_{5}\right)$	3.39 ± 1.39	3.44 ± 1.42	ns	4.11 ± 1.61	3.66 ± 1.99	ns
$BW(f_{0})$ (Hz)	1.20 ± 0.73	1.02 ± 0.39	ns	1.27 ± 1.21	2.33 ± 5.90	ns
$BW(f_{1})$ (Hz)	0.95 ± 0.16	0.87 ± 0.11	ns	0.93 ± 0.15	0.90 ± 0.11	ns
OI	0.45 ± 0.10	0.54 ± 0.11	0.003	0.48 ± 0.09	0.51 ± 0.06	ns
LK	0.83 ± 0.05	0.91 ± 0.04	<0.001	0.85 ± 0.06	0.89 ± 0.04	0.011

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
