# Peer review of "Spectral Analysis and Mutual Information Estimation of Left and Right Intracardiac Electrograms during Ventricular Fibrillation"

_sensors, 2020, doi:10.3390/s20154162_

Round 1
Reviewer 1 Report
The manuscript describes a study of intracardiac electrogram signals from 22 patients upon induced ventricular fibrillation and indication for CHF. The work includes an extensive study of the relationship between RV and LV signals, including spectral and information theoretical analyses. The work is very interesting and well-motivated considering the current literature. However, some details on how the experiments were conducted are not sufficiently clear. The manuscript would also benefit from a deeper discussion of the potential impact of this work in ventricular fibrillation diagnosis and automatic recognition, for example.
Strengths: (1) novel study; (2) extensive experiments; and (3) very well motivated;
Weaknesses: (1) some unclear experimental details; (2) not clear what is the potential impact of this study.
Comments:
- The abstract is almost 400 words long. It should be shorter and straight to the point, especially the part about the results and conclusions (only the very few main ones should be included in the abstract).
- I understand the novelty of the study conducted by the authors, but it is not clear which are the real contributions of such study. How would the conclusions of this work help, for example, VF diagnosis or automatic detection methods? What are the potential impact and consequences of this study? This should be clearer in the introduction and in the conclusion.
- Regarding the acquired data, it would be relevant for the sake of reproducibility for the acquired signals to be available (there are many examples of such data being made publicly available in platforms like Physionet, for example). This would also be a great contribution, if the authors are able/allowed to do it.
- Computing information theory measurements like mutual information is not an easy task in these contexts. For multi-variate data like signals with several quantization levels this is a very lengthy and difficult task and the most common solution is to use approximations. The authors should describe better how they compute these measurements. Making the code available would be a great step towards reproducibility.
- In Figures 7 and 8, using a line plot makes it appear like there is a connection between points along the x-axis, but it relates to different subjects. Maybe using just a scatter plot would make the figures clearer.
- Page 4, line 133, EMG -> EGM.
Reviewer 2 Report
The Authors introduced spectral and information theory analyses for simultaneously acquired LV and RV signals during induced VF in subjects with CHF and clinical indication for BICD implantation.
The Authors correctly pointed out differences in sensing setup for LV and RV. Are they sure that the differences reported in Table 1 aren't due to this difference in sensing configurations? In case, they should discuss this point in depth and provide sufficient evidence that differences are not due to confounding factors.
When comparing entropy, they state that entropy in the LV is more variable than in RV: can they provide some metrics for the assessment of this variability?
How did they get p-values? I expect they used a paired test, but it should be described in Methods.
A few minors:
- English spelling should be checked
- Abbreviations are sometimes mispelled (e.g.: VL instead of LV, etc...)
- Some phrases are incomplete (e.g.: line 73)
Reviewer 3 Report
The authors study the properties of EGM recordings of patients during a testing phase of the defibrillator while it is being implanted. The properties studied can be classified in two classes, spectral and entropy-based. The authors present the results obtained on a small group of subjects and relate them to the state of the art in the field.
My first critique of the article is the language. There are just too many typos for the article to be accepted as it is. Below, I present just the problems that I found after one scan of the text:
- line 73, missing sentence end: Section 2 summarizes the .
- line 133, typo: A typical EMG signal...
- line 133, typo: which is commonly knows...
- line 146, rethink the word grosser here: the first few seconds can be slightly grosser...
- line 147, reduce compared to what? ...tend to reduce the voltage...
- line 183, probably not disconnected, or it would not have a function: wire to the atrium is kept disconnected at all times
- throughout the text VR instead of RV and sometimes VL instead of LV
- line 194, 'digitation' is incorrect in this context: The final result of the digitation...
- line 197, I do not understand the math here: Forty VF recordings were collected from 22 patients with clinical indication for a biventricular cardioverter defibrillator, with one recording per patient included in the study.
- line 260, Is the statistics in the parenthesis only for men or for all the patients? Twenty-two patients (15 men, 63.2 ±11.5 years old) ...
- (not exactly a language problem, so forgive me to include it here) I had to dig in the literature to discover why would "VF be induced" in patients. I would recommend that such sentences were given more context, the article is being submitted to section "Biomedical Sensors", where not all readers will be experts in defibrillators.
The second critique is regarding the treatment of the two spectral and the entropy-based analyses. The latter is elaborated in far greater depth than the former. The latter also has a theoretical example added, which I struggled with at first (as it is included in the section Results, while it is clearly not a result itself), but I find it very helpful to be there. Spectral analyses is presented very briefly in comparison but it probably should not be. The results are largely statistically significant, which goes against my intuition, and that is a good thing. For example, I would expect the results to be more noisy and more dependent on some hidden variables / immeasurable properties of the heart (e.g. myocardium volume). And for an example of what could be added to give the section depth is: what is the significance of dominant and fundamental frequencies? Is the dominant frequency not susceptible to noise in this example - could it be that two neighbouring peaks are very close in amplitude and the dominant one appears dominant by pure chance? Etc....
Lastly, I struggle to justify the placement of this article into journal Sensors, since it only presents the analysis of measurements. It has very little connection to actual sensors. The authors even had to digitize EMG recordings, while the type of sensor used suggests digitization at the point of acquisition and storage. Therefore I suggest the authors to make a bigger effort to connect their work with the world of sensors.
Round 2
Reviewer 1 Report
I believe the authors have answered all my comments and adequately revised the manuscript. I wish to thank them for doing so and to congratulate them on a good research work.
Reviewer 2 Report
All the recommendations have been fulfilled.
I have no further comments.